# Potential Targets for Deprescribing in Medically Complex Older Adults with Suspected Cognitive Impairment

**DOI:** 10.3390/geriatrics7030059

**Published:** 2022-05-19

**Authors:** Juliessa M. Pavon, Theodore S. Z. Berkowitz, Valerie A. Smith, Jaime M. Hughes, Anna Hung, Susan N. Hastings

**Affiliations:** 1Department of Medicine/Division of Geriatrics, Duke University, Durham, NC 27710, USA; susan.hastings@duke.edu; 2Geriatric Research Education Clinical Center, Durham Veteran Affairs Health Care System, Durham, NC 27705, USA; 3Claude D. Pepper Center, Duke University, Durham, NC 27710, USA; 4Health Services Research & Development, Durham Veterans Affairs Health Care System, Durham, NC 27701, USA; theodore.berkowitz@va.gov (T.S.Z.B.); valerie.smith9@va.gov (V.A.S.); jhughes@wakehealth.edu (J.M.H.); 5Department of Population Health Sciences, Duke University, Durham, NC 27701, USA; anna.hung@duke.edu; 6Department of Medicine/Division of General Internal Medicine, Duke University, Durham, NC 27710, USA; 7Department of Implementation Science, Wake Forest School of Medicine, Winston-Salem, NC 27157, USA; 8Section on Gerontology and Geriatric Medicine, Division of Public Health Sciences, Department of Internal Medicine, Wake Forest School of Medicine, Winston-Salem, NC 27103, USA

**Keywords:** deprescribing, anticholinergic burden, medication side effects, cognitive impairment, medical complexity

## Abstract

Deprescribing may be particularly beneficial in patients with medical complexity and suspected cognitive impairment (CI). We describe central nervous system (CNS) medication use and side effects in this population and explore the relationship between anticholinergic burden and sleep. We conducted a cross-sectional analysis of baseline data from a pilot randomized-controlled trial in older adult veterans with medical complexity (Care Assessment Need score > 90), and suspected CI (Telephone Interview for Cognitive Status score 20–31). CNS medication classes included antipsychotics, benzodiazepines, H2-receptor antagonists, hypnotics, opioids, and skeletal muscle relaxants. We also coded anticholinergic-active medications according to their Anticholinergic Cognitive Burden (ACB) score. Other measures included self-reported medication side effects and the Pittsburgh Sleep Quality Index (PSQI). ACB association with sleep (PSQI) was examined using adjusted linear regression. In this sample (N = 40), the mean number of prescribed CNS medications was 2.2 (SD 1.5), 65% experienced ≥ 1 side effect, and 50% had an ACB score ≥ 3 (high anticholinergic exposure). The ACB score ≥ 3 compared to ACB < 3 was not significantly associated with PSQI scores (avg diff in score = −0.1, 95% CI −2.1, 1.8). Although results did not demonstrate a clear relationship with worsened sleep, significant side effects and anticholinergic burden support the deprescribing need in this population.

## 1. Introduction

Deprescribing—a systematic process of identifying medications to stop, taper, or substitute—has emerged as a critical consideration in high-quality medication management, especially among older adults [1]. Deprescribing interventions that aim to reduce central nervous system (CNS) medication use, especially the use of medications with anticholinergic activity, such as tricyclic antidepressants and muscle relaxants [2,3,4], may be particularly beneficial for medically complex older adults with suspected cognitive impairment. Patients with medical complexity (measured by high hospitalization risk) often have multiple chronic conditions, exhibit patterns of uncoordinated health care utilization [5,6,7], and are at further risk for unrecognized cognitive impairment (CI) [8,9]. There are many contributors to cognitive impairment in this population, not only normative cognitive decline, but also factors that affect cognition in the short term, such as medications. While there is significant evidence demonstrating the negative impact of anticholinergic medications on cognition, physical function [10], and side effects (such as blurred vision, constipation, dry mouth, falls) [11], the association with sleep quality remains overlooked [12]. Poor sleep quality may contribute to increased fatigue and reduced daytime functioning [13]. As a result, the use of medications with anticholinergic activity may further reduce patient capacity to manage illness and treatment burden [13,14,15]. This reduced capacity may be particularly consequential in older adults with suspected cognitive impairment.

Deprescribing CNS medications or potentially inappropriate medications, such as anticholinergics, can be an opportunity to improve care and important patient-reported outcomes in this population [16,17,18]. However, the outcomes for deprescribing may vary by medication classes and the number of medications. Thus, it remains uncertain which medication classes to target for the highest impact. Here we explored potential targets for deprescribing in medically complex older adults with suspected cognitive impairment by (1) describing CNS medication use and common side effects in medically complex patients with suspected CI, (2) identifying the medications contributing to anticholinergic cognitive burden (ACB), which may increase the risk for future cognitive decline, and (3) examining the relationships between ACB and sleep. Our study is among the first to explicitly examine this relationship in a high-risk population of older adult veterans, and we hypothesized that sleep quality is worse among those with higher anticholinergic burden.

## 2. Methods

### 2.1. Participants and Setting

This is a cross-sectional secondary analysis of a pilot randomized control trial of community-dwelling older adults, aged 65 years or greater, enrolled in Durham Veterans Affairs Health Care System-affiliated primary care clinics, with medical complexity and suspected cognitive impairment [19]. The purpose of this pilot trial was to test the feasibility, acceptability, usability, and perceived value of a 14-week video-delivered nurse care management program for medically complex older veterans with CI and their care partners compared with a similar telephone-based program. Medical complexity was defined as having a Care Assessment Need (CAN) score > 90, a multi-variable score predicting hospitalization risk within 1 year. CAN score is expressed as a percentile, such that these are people identified to be in the top 10% likelihood of hospitalization within 1 year [7]. Patients who had previously recognized cognitive impairment (e.g., if medical records indicated known dementia or cognitive impairment (CI) or lack of decision-making capacity) were excluded. Potentially eligible participants were screened for cognitive impairment using the modified version of the Telephone Interview for Cognitive Status (TICS-m) [20]. Those with scores between 20 and 31 (education-adjusted), a range previously correlated with mild CI, were offered enrollment. Other exclusion criteria included serious mental illness (mental, behavioral, or emotional disorder resulting in serious functional impairment) or active substance abuse, currently hospitalized or residing in a nursing home or other institutional care setting, hospice-eligible, or unable to communicate by telephone. 

### 2.2. Measures

#### 2.2.1. Medication Data

Medication data were collected during the first of three scheduled study nurse calls with patients (considered study baseline) and were derived from a reconciled medication list generated by a study nurse using the patient report and the electronic health record of the patient’s current medications. Medications in the following central nervous system (CNS) medication classes were recorded: antipsychotics, benzodiazepines, H2-receptor antagonists, hypnotics, opioids, and skeletal muscle relaxants. For all medications, we also identified and coded anticholinergic-active medications according to their score on the 2012 update of the Anticholinergic Cognitive Burden (ACB) scale: ACB1, possibly anticholinergic; ACB2, definitely anticholinergic; ACB3 definitely anticholinergic and associated with delirium [21]. All other drugs were assigned a score of 0 (ACB0). ACB scores were summed for each patient and then further categorized as no/low exposure to anticholinergic medications (total ACB score = 0–2) vs. high exposure (total ACB score 3+) [11,22,23]. 

Of note, medications in the original study were selected to meet the study aims, which was to better target care management. Thus, medication classes were included because they are either medications that are (a) included as an anticholinergic medication in the ACB scale, (b) medication classes that can impact cognition, or (c) both. For this analysis, we included all original study medications because of their potential to impact patient outcomes.

#### 2.2.2. Side Effects and Sleep

Patients were asked by study nurses to self-report if they had any of the following symptoms or problems: blurry vision, constipation, dizziness, dry mouth, dry eyes, sleepiness, difficulty emptying the bladder, and falls. Patients could indicate having one or more of these symptoms. The Pittsburgh Sleep Quality Index (PSQI), an 18-item self-report questionnaire, (0–21, scores ≥ 5 indicate poor sleep quality) is the gold standard self-report measure to assess sleep disturbance over the prior month [24]. The items produce seven component scores which range from 0 (no difficulty) to 3 (severe difficulty): sleep duration, sleep disturbance, sleep latency, daytime disturbance, habitual sleep efficiency, sleep quality, and use of sleep medications. All self-reported measures were also collected at baseline for all participants.

#### 2.2.3. Analysis

Descriptive statistics were calculated for all study variables. Distributions of demographics, TICS-m, side effects, and PSQI scores measures were calculated for all patients, as well as separately for those with ACB score 0–2 versus 3+. Multivariable linear regression models were used to determine the relationship between ACB score (0–2 vs. 3+) and sleep (PSQI) adjusted for age and race. Analyses were performed using SAS, version 9.0 (SAS Institute, Inc., Cary, NC, USA).

## 3. Results

### 3.1. CNS Medication Burden and Side Effects

Among 40 community-dwelling veterans with medical complexity and suspected CI (mean age 72.4 (SD 6.1), 100% male, 37.5% Black), the mean number of prescribed medications from the included classes was 2.2 (SD 1.5). Overall, 73% were taking anticholinergic-active medications (*n* = 29). Among the CNS medication classes, the top two most commonly prescribed were opioids (*n* = 13), and skeletal muscle relaxants (*n* = 7) (Table 1). Further, 65% of all veterans taking the included medication classes experienced one or more side effects (Table 1), with more than a quarter of the sample experiencing any one of the following: dry mouth (38%), sleepiness (33%), dry eyes (25%), and dizziness (25%).

### 3.2. Most Frequently Prescribed Medications with Anticholinergic Activity 

The prescribed medication classes with anticholinergic activity are presented in Figure 1, and the medication names are listed in Table 2. The two most frequently prescribed medication classes with ACB score 3 were systemic antihistamines (20% of sample) and skeletal muscle relaxants (8%). The four most frequently prescribed medications were diphenhydramine (antihistamine), methocarbamol (muscle relaxant), hydroxyzine (antihistamine), and oxybutynin (antimuscarinic), with diphenhydramine or methocarbamol prescribed to more than 20% of the sample, and to 40% of those who had a total ACB score ≥ 3.

The most frequently prescribed medication class with ACB score 2 was skeletal muscle relaxant (10% of the sample), with 100% of those receiving cyclobenzaprine. Prescribed medications with ACB score 1 belonged to a diverse range of medication classes. The majority of patients with prescriptions for medications with ACB score 1 received beta-blockers (metoprolol) (40% of sample), followed by diuretics (furosemide) (18%), systemic antihistamines (cetirizine, loratadine, desloratadine) (13%), and antithrombotics (warfarin) (10%) (Figure 1 and Table 2). 

### 3.3. Anticholinergic Burden

Approximately three out of four study participants (73%) were prescribed medications with anticholinergic activity and exposed to anticholinergic burden (total ACB score ≥ 1). Mean total ACB score at baseline was 2.6 (SD 2.1), and 50% had a total ACB score ≥ 3 (*n* = 20) (high burden). Characteristics of participants with a total ACB score ≥ 3 and those with a total ACB score < 3 are shown in Table 1. The most frequently prescribed individual medications to patients with a total ACB score ≥ 3 were metoprolol (beta-blocker, ACB = 1), furosemide (diuretic, ACB = 1) (combined represents 75% of patients with a total ACB score ≥ 3), or diphenhydramine (systemic antihistamine) (Table 2). 

### 3.4. Total ACB and Association with Sleep

The mean PSQI score was 9. Overall, 95% of participants had a PSQI score of ≥5 which indicates poor sleep quality. The ACB score was not significantly associated with PSQI scores (average difference in score between veterans with ACB score ≥ 3 vs. scores < 3 = −0.1, 95% CI −2.1, 1.8).

## 4. Discussion

One-half of veterans with medical complexity and suspected cognitive impairment had a high anticholinergic burden, 2 out of 3 reported side effects attributable to anticholinergic medications, and the majority had worse than average sleep quality rating compared to the general population of a similar age [25]. Although results do not demonstrate a clear relationship with worsened sleep as hypothesized, significant side effect burden, overall poor sleep quality, and other known adverse effects of CNS and anticholinergic medications support the need for deprescribing in this population.

In our study, 50% of medically complex veterans with suspected cognitive impairment are exposed to the high anticholinergic burden, which is similar to proportions seen in other older adult populations, but nearly double that seen in cognitively impaired populations [26,27]. This is a troubling result given the associations between anticholinergic burden, including cumulative burden, and limitations in cognitive and physical function [3,4,10]. Studies have also revealed longitudinal impacts on dementia and poor physical functioning [28,29,30]. However, many of these studies examine the overall burden of anticholinergics and sedative CNS medications, but the individual impact of target medications is not known. Further, there is also concern about the potential misuse of these medications [31]. In our study, diphenhydramine (ACB of 3) was one of the most frequently prescribed medications in this group. According to the 2019 Beers Criteria for Potentially Inappropriate Medication Use in Older Adults, individuals 65 years or older should avoid the use of diphenhydramine [32]; however, many over-the-counter sleep products contain diphenhydramine. In a group where poor sleep quality is common, the potentially inappropriate use of diphenhydramine is likely to be high. Methocarbamol, another frequently prescribed medicine with an ACB of 3, also produces more pronounced anticholinergic side effects in older adults, such as drowsiness, dizziness, and falls, and it should be prescribed with caution [33]. In the context that our findings do not really support using total ACB score as a targeting tool for deprescribing, particularly given that some of the most common medications among those with high total ACB scores are unlikely to contribute to anticholinergic side effects (e.g., metoprolol and furosemide), individual medications with high anticholinergic activity do become important targets for deprescribing interventions, especially those prescribed without a compelling clinical indication.

One important consideration with deprescribing is identifying the right timing and setting. Deprescribing interventions aimed at reducing acute care utilization among patients with medical complexity is an example. High-risk medication may reduce overall capacity (e.g., sleep, physical activity, cognition) in medically-complex older adults with mild cognitive impairment [10,12,13,14,15], which can negatively impact the ability to manage acute health stressors, thereby increasing acute care utilization [15]. Thus, there is a need to look for opportunities aimed at reducing acute care utilization among patients with suspected cognitive impairment. Deprescribing among those with high risk for hospitalization, not scoring well on CI screening, and who have a high anticholinergic burden, for example, could be an opportunity to intervene and improve care.

Our results also cast a new light on the side effect burden among medically complex older adults. The accumulation of side effects can promote prescribing cascades, where additional medications are prescribed to treat the adverse side effects of previously prescribed medications [34]. This can lead to polypharmacy and further increases the risk for adverse drug events and future hospitalization [35]. Telehealth care management strategies, for example, can provide systematic opportunities for identifying and managing CNS medication-related side effects [19]. These side effects can be important targets for deprescribing strategies aimed at mitigating the risk for poor outcomes in medically complex patients with suspected cognitive impairment.

Limitations include involvement of a single site, male-only enrollment, and small sample size which limits generalizability and power to detect small effect sizes, and the inability to accurately determine if a veteran was taking a medication prescribed through a non-VA provider. However, this was mitigated by using a reconciled medication list using both patient reports and the electronic health record. Although comorbidity information is not available, baseline medical complexity CAN scores, Generalized Anxiety Disorder-7, and Patient Health Questionnaire-9 scores for this sample are reported elsewhere [19]. Further, self-reported side effects relied on patient recall, such that our data likely underestimates medication-related side effects. 

## 5. Conclusions

Older veterans with medical complexity and suspected cognitive impairment have a high anticholinergic burden, as well as significant side effects attributed to medications with anticholinergic activity. A process to reduce anticholinergic-related medications, such as deprescribing interventions and care management, is warranted. Future studies in this population assessing the longitudinal impact of an anticholinergic burden on cognition, physical activity, and sleep is needed to further support deprescribing efforts.

## Figures and Tables

**Figure 1 geriatrics-07-00059-f001:**
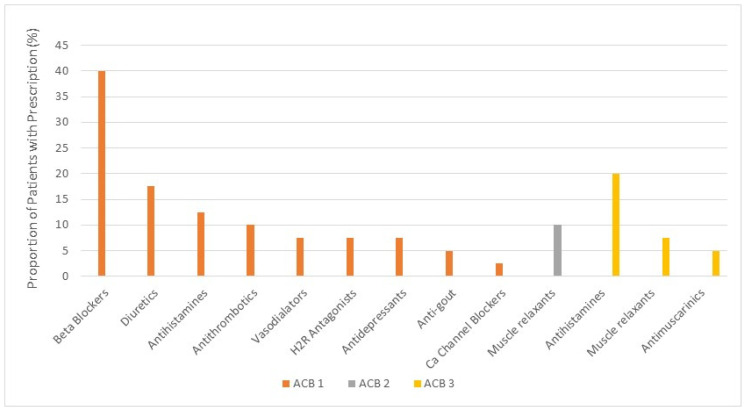
Most frequently prescribed medication classes by anticholinergic cognitive burden score. ACB = anticholinergic cognitive burden; H2R = histamine type 2 receptor. antihistamine ACB 1 = cetirizine, loratadine, desloratadine; antihistamine ACB 3 = diphenhydramine, hydroxyzine; muscle relaxant ACB 2 = cyclobenzaprine; muscle relaxant ACB 3 = methocarbamol.

**Table 1 geriatrics-07-00059-t001:** Baseline patient characteristics and clinical measures by total anticholinergic cognitive burden score.

Veteran Characteristic at Baseline	Total Sample N = 40	ACB 0–2 *n* = 20	ACB 3+ *n* = 20
Age, mean (SD)	72.4 (6.1)	70.5 (5.0)	74.3 (6.5)
Male gender, n (%)	40 (100.0)	20 (100.0)	20 (100.0)
Black or African-American race, n (%)	15 (37.5)	9 (45.0)	6 (30.0)
Married or living together, n (%)	31 (77.5)	16 (80.0)	15 (75.0)
High school education or less, n (%)	14 (35.0)	10 (50.0)	4 (20.0)
Fair/poor self-rated health, n (%)	26 (65.0)	15 (75.0)	11 (55.0)
Taking anticholinergic-active medication ^a^, n (%)	29 (72.5)	12 (60.0)	17 (85.0)
Taking other prescribed CNS medications, n (%)			
Antipsychotics	0	0	0
Benzodiazepines	0	0	0
H2-Receptor antagonists	3 (7.5)	1 (5.0)	2 (10.0)
Hypnotic sedatives	2 (5.0)	1 (5.0)	1 (5.0)
Opioids	13 (32.5)	6 (30.0)	7 (35.0)
Skeletal muscle relaxants	7 (17.5)	0	7 (35.0)
Any side effects, n (%)	26 (65.0)	12 (60.0)	14 (70.0)
TICS-m score, education adjusted, mean (SD)	27.6 (2.6)	26.9 (2.6)	28.3 (2.5)
PSQI/sleep, mean (SD)	9.1 (2.8)	9.3 (2.7)	9.0 (2.9)

^a^ The designation of “anticholinergic” was not exclusive to the other classifications, e.g., ranitidine was categorized as both an anticholinergic and an H2-receptor antagonist, and cyclobenzaprine was categorized as both an anticholinergic and skeletal muscle relaxant. ACB = anticholinergic cognitive burden; CNS = central nervous system; H2 = histamine type 2; TICS-m = Telephone Interview for Cognitive Status—modified; PSQI = Pittsburgh Sleep Quality Index.

**Table 2 geriatrics-07-00059-t002:** Frequency of all prescribed medications, overall and in patients with low vs. high anticholinergic burden.

ACB Scale Value	Class(es)	Medication	Number (%) Patients (N = 40)	Number (%) Patients (ACB 0–2) (*n* = 20)	Number (%) Patients (ACB 3+) (*n* = 20)
0	Opioid	OXYCODONE	9 (22.5)	4 (20.0)	5 (25.0)
0	Opioid	TRAMADOL	4 (10.0)	1 (5.0)	3 (15.0)
0	Hypnotic/sedative	ZOLPIDEM	2 (5.0)	1 (5.0)	1 (5.0)
0	Opioid	HYDROCODONE	1 (2.5)	1 (5.0)	0
1	Anticholinergic-active	METOPROLOL	16 (40.0)	7 (35.0)	9 (45.0)
1	Anticholinergic-active	FUROSEMIDE	7 (17.5)	1 (5.0)	6 (30.0)
1	Anticholinergic-active	WARFARIN	4 (10.0)	2 (10.0)	2 (10.0)
1	Anticholinergic-active	CETIRIZINE	3 (7.5)	1 (5.0)	2 (10.0)
1	Anticholinergic-active AND H2-receptor antagonist	RANITIDINE	3 (7.5)	1 (5.0)	2 (10.0)
1	Anticholinergic-active	COLCHICINE	2 (5.0)	1 (5.0)	1 (5.0)
1	Anticholinergic-active	ISOSORBIDE	2 (5.0)	0	2 (10.0)
1	Anticholinergic-active	TRAZODONE	2 (5.0)	0	2 (10.0)
1	Anticholinergic-active	BUPROPION	1 (2.5)	1 (5.0)	0
1	Anticholinergic-active	DESLORATADINE	1 (2.5)	0	1 (5.0)
1	Anticholinergic-active	DIGOXIN	1 (2.5)	0	1 (5.0)
1	Anticholinergic-active	HYDRALAZINE	1 (2.5)	0	1 (5.0)
1	Anticholinergic-active	LORATADINE	1 (2.5)	0	1 (5.0)
2	Anticholinergic-active AND skeletal muscle relaxant	CYCLOBENZAPRINE	4 (10.0)	0	4 (20.0)
3	Anticholinergic-active	DIPHENHYDRAMINE	5 (12.5)	0	5 (25.0)
3	Anticholinergic-active AND skeletal muscle relaxant	METHOCARBAMOL	3 (7.5)	0	3 (15.0)
3	Anticholinergic-active	HYDROXYZINE	2 (5.0)	0	2 (10.0)
3	Anticholinergic-active	OXYBUTYNIN	2 (5.0)	0	2 (10.0)
3	Anticholinergic-active	DOXYLAMINE	1 (2.5)	0	1 (5.0)

ACB = anticholinergic cognitive burden.

## Data Availability

Data available on request due to restrictions, e.g., privacy or ethical concerns.

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
