# Peer review of "Potential Targets for Deprescribing in Medically Complex Older Adults with Suspected Cognitive Impairment"

_geriatrics, 2022, doi:10.3390/geriatrics7030059_

Round 1

Reviewer 1 Report

Thank you for the opportunity to review this manuscript.  I think this study would be of interest to the readership of this journal.  There are however a number of edits that I would suggest in order to improve clarity and interpretation for readers.  Please see my comments below.

ABSTRACT

Description of study design

  • The abstract itself is lacking in detail regarding the makeup of the sample and ascertainment of outcomes, namely medication-related adverse effects, which are not described. Please consider adding if word limits will allow.

Introduction

Lines 41-59

  • The opening paragraph of the introduction is quite lengthy and repetitive in places. Please review and consider scaling back to be more concise.

Lines 46-47 – “Overall, these patients have high medical burden and limited capacity to manage complicated regimens.”

  • Suggest softening language to “reduced capacity”

Lines 50-51 – “There are many reasons for cognitive impairment in this population…”

  • Suggest changing “reasons” to “contributors”

Lines 58-60 – “As a result, use of medications with anticholinergic activity among medically complex older adults may influence patient capacity (cognitive impairment or poor sleep) to manage illness and treatment burden.”

  • I don’t know that the focus on cognitive impairment and sleep as outcomes is well-justified here. In the prior statement, the authors mention a whole host of negative outcomes, including physical function, which is not addressed at all. 
  • It seems like the focus on sleep and cognition is more so out of convenience because these measurements were available. If that is the case, then a stronger case may need to be made for focusing on cognitive function and sleep quality specifically.  I would suggest providing a bit more detail from the literature referenced in the prior sentences regarding the potential association of anticholinergic medication use with these outcomes.

Lines 62-63 – “However, it remains uncertain which medication classes to target for highest impact.”

  • It would be pertinent to include some literature on studies that have attempted to deprescribe CNS medications and PIMs broadly and their limitations. Then tie in why it might be pertinent to develop a more targeted approach - i.e. the rationale for this study.

METHODS

Subjects and setting – Lines 71-84

  • A bit more detail would be helpful for reviewers to understand the make up of the sample. Suggest addressing where were subjects recruited from. (Just a simple "from an outpatient clinic" would suffice.)
  • Also would suggest a brief description of the original aims of the trial so the reader can understand what measurements were available.

Cognitive Screening – Lines 78-81

  • Suggest clarifying what a score of 20-30 on the TICS-m corresponds to from a clinical point of view. So to clarify, these are patients with newly identified or suspected cognitive impairment and not confirmed, right?
  • A point worth consideration is the fact that the limited range of scores here may be a contributor to the negative results observed. e., is there enough variability in cognitive screening scores to detect a difference by medication use? Probably not given that all subjects had scores in the range of 20-30.  Might be a reason to not evaluate cognition as an outcome.

Medication data – Lines 87-99

  • The classification of medications in the study is a bit confusing, especially in the results when flipping back and forth between ACB and CNS medications. Why the focus on both?
  • Were ACB scores assigned to all medications? Or just those that were in the classes of CNS medications mentioned?
  • It seems like most of the big offenders for CNS medications would be included in the ACB, right? So would suggest just focusing on those meds included in the ACB, which can still be reported at the class level.

Side effects and sleep – Lines 100-107

  • I assume cognitive function based on the screening assessment? Would be pertinent to mention that again here and how it was operationalized, given that the range of scores was so limited based on the inclusion criteria (as I mention above).

Analysis – Lines 108-114

  • In the introduction, there is quite a bit of emphasis placed on 'medically complex' older adults, yet the medical complexity of those individuals included in this study is not described or mentioned. I assume medical complexity was not measured as part of the study?  I see that PHQ-9 was measured.  Why was this not incorporated as a covariate?
  • Suggest commenting on this in the discussion.

RESULTS

Lines 117-119 – “Among 40 community-dwelling Veterans with medical complexity…”

  • This is the first time that the reader is aware that the study focuses on a Veteran population. Suggest mentioning this earlier in the methods.
  • Additionally, there is no data reported in the methods or in the sample characteristics to indicate that the study sample has medical complexity. This needs to be addressed somewhere in the methods or else reported in the results, particularly given the emphasis in the introduction and discussion on ‘medical complexity’.

Lines 160-164 – “The most frequently prescribed individual medications to patients with total ACB score ≥3 were…”

  • Here it would be pertinent to point out that furosemide and metoprolol both have a score of 1.

DISCUSSION

Lines 194-196 – “In the context that our findings do not really support using total ACB score as a targeting tool for deprescribing…”

  • Suggest adding something to the effect of… particularly given that some of the most common medications among those with high ACB scores are unlikely to contribute to anticholinergic side effects (e.g., metoprolol and furosemide).

Lines 230-232 – “Future studies assessing the longitudinal impact of anticholinergic burden on cognition, physical activity, and sleep is needed to further support deprescribing efforts.”

  • There are actually a number of studies that have examined the longitudinal impact of anticholinergic medication use on cognition and physical activity. Suggest adding a discussion of those to the discussion section.

Reviewer 2 Report

Thanks for recommending me as a reviewer. In this paper, author aim to describe central nervous system (CNS) medication use and side effects in this population and explore relationships between anticholinergic burden and cognitive function and sleep. authors conducted a cross-sectional analysis of baseline data from a pilot randomized-controlled trial. CNS medication classes included antipsychotics, benzodiazepines, H2-receptor antagonists, hypnotics, opioids, and skeletal muscle relaxants. authors also coded anticholinergic-active medications according to their Anticholinergic Cognitive Burden (ACB) score. If the authors complete minor revisions, the quality of the study will be further improved.

  1. The introduction is well written. If the authors provide a richer theoretical background in the introduction section, it can help readers understand.
  2. Authors need to further describe the discussion section. Authors can refer to more research in the Discussion section.
  3. In the discussion section, authors can add more limitations.

Reviewer 3 Report

Thank you very much for sending me this manuscript for review. It is very interesting and you can see that the authors of the study have made a great effort. Below are a number of considerations:

  • It is recommended to increase the introduction so that the "state-of-the-art" is more complete and developed. The information that has been used is adequate; it would be a matter of expanding it. 
  • The last paragraph of the introduction shows the objectives and proposes results, i.e. hypotheses. Please separate and list them to make clear the objectives on the one hand, and the hypotheses pursued on the other (e.g., h1, h2, etc.). 
  • As for the description of the subjects, what was the mean age and standard deviation? And the percentage of women and men? Describe it in this subsection. 
  • Value the possibility of naming Participants instead of Subjects although both concepts can be used. 
  • Regarding the scales/instruments used, is there any further evidence to support their use? Would it be possible to locate an internal consistency index or coefficient or similar? Please describe the measures in more detail, giving examples of items and description of scales where appropriate.
  • The sample used is small. Please justify the number used and whether it is feasible to perform the statistical analysis.
  • The hypotheses formulated should be cited with the numbering used in the introduction (e.g., h1, h2, etc.). The rest of the discussion is adequate, raises limitations and also future lines of research.
  • The conclusions are adequate and describe the main results of the study.

Thank you very much for your work. It is a very interesting research. 
